# Geographic Distribution of Avirulence Genes of the Rice Blast Fungus *Magnaporthe oryzae* in the Philippines

**DOI:** 10.3390/microorganisms7010023

**Published:** 2019-01-19

**Authors:** Ana Liza C. Lopez, Tapani Yli-Matilla, Christian Joseph R. Cumagun

**Affiliations:** 1Jose Rizal Memorial State University–Tampilisan Campus, Znac, 7116 Tampilisan, Zamboanga del Norte, Philippines; analiza.5631@gmail.com; 2Institute of Weed Science, Entomology and Plant Pathology, College of Agriculture and Food Science, University of the Philippines Los Baños, 4031 Los Baños, Laguna, Philippines; 3Molecular Plant Biology, Department of Biochemistry, University of Turku, FI-20520 Turku, Finland; tymat@utu.fi; 4Molecular Phytopathology and Mycotoxin Research, University of Göttingen, 37077 Göttingen, Germany

**Keywords:** avirulence genes, *Magnaporthe oryzae*, Philippines, resistance genes, rice blast

## Abstract

A total of 131 contemporary and 33 reference isolates representing a number of multi-locus genotypes of *Magnaporthe oryzae* were subjected to a PCR test to detect the presence/absence of avirulence (*Avr*) genes. Results revealed that the more frequently occurring genes were *Avr-Pik* (81.50%), *Avr-Pita* (64.16%) and *Avr-Pii* (47.98%), whereas the less frequently occurring genes were *Avr-Pizt* (19.08%) and *Avr-Pia* (5.20%). It was also laid out that the presence of *Avr* genes in *M. oryzae* is strongly associated with agroecosystems where the complementary resistant (*R*) genes exist. No significant association, however, was noted on the functional *Avr* genes and the major geographic locations. Furthermore, it was identified that the upland varieties locally known as “Milagrosa” and “Waray” contained all the *R* genes complementary to the *Avr* genes tested.

## 1. Introduction

The interaction between rice and the rice blast fungus, *Magnaporthe oryzae*, is a well-documented plant pathosystem. Valent [1] reported rice blast as a model system to further one’s understanding on the concept of host species specificity and cultivar specificity, including the mechanisms of plant pathogenesis. The rice-blast pathogen interaction fits the gene-for-gene system, and resistance in rice cultivars are due to specific interactions between *R* genes in a host and *Avr* genes of *M. oryzae* strains under favorable conditions [2,3]. *Magnaporthe oryzae* can easily mutate to overcome such specific interactions, thereby challenging plant pathologists and plant breeders for its effective management.

Avirulence gene in plant pathogens is simply defined as a gene encoding for a specific protein which is recognized by a host plant bearing the matching *R* gene [4]. The interaction between the *Avr* gene and its corresponding *R* gene follows the classic gene-for-gene concept as illustrated by reference [5]. The primary or secondary products of *Avr* genes, termed as “elicitors” induces the host receptor to mount various defense responses which oftentimes involve a hypersensitive response [6]. Fungal *Avr* genes have been successfully isolated by reverse genetics and positional cloning [6]. To date, several *Avr* genes had been recognized in plant pathogens [7,8]. The cloned avirulence genes identified in *M. oryzae* include the four genes of *PWL* [9,10], *AVR-Pita* [11], *ACE1* [12], *AvrPiz-t* [13], *Avr-Pia* [14,15], *Avr-Pii* [15], *Avr-Pik/km/kp* [15], *Avr1-CO39* [16] and *AvrPi9* [17]. Among these, *Avr*-*Pita* and *Avr-Pii* were linked to a subtelomeric location, suggesting that loss of chromosome tips could result in gain of virulence of *M. oryzae* [11,18]. On the other hand, Sone et al. [19] reported homologous recombination, which causes DNA rearrangements, deletions, translocations and even horizontal transfers between strains, as a mechanism for the loss of *Avr-Pia* and adaptation of *M. oryzae* against resistant rice cultivars. In addition, the presence of transposable elements such as *MAGGY* and *Pot3* adjacent to the gene regions of *Avr*s- *Pita*, *Pii*, *Pia* and *Pizt* aid the pathogen in overcoming host *R* genes, making the *Avr* genes gained or lost during the evolution of the pathogen [15,20]. Moreover, nucleotide substitutions were noted in *Avr-Pik* polymorphisms [21]. It was further cited that, in order for *M. oryzae* to adapt to rice *R* genes, it must have to lose or recover its *Avr* genes [15,19,21,22]. These scenarios constitute an arms race between the *R* gene and the pathogen *Avr* gene. Selection pressure on the host plant population favors direct or indirect recognition of the pathogen avirulence genes through its *R* gene to mount defense responses, whereas selection pressure on the pathogen population favors escape of host recognition [23]. Recently, host jumps were observed in the case of *M. oryzae* isolated from ryegrass, which cannot infect wheat due to recognition of the avirulence gene *PWT3* in the ryegrass pathogen by the resistance gene *Rwt3* in wheat. The loss of the avirulence gene resulted in wheat infection of the ryegrass pathogen, when *PWT3* was disrupted [24]. In our previous study, we tested the presence of *ACE1* genotypes among 53 Philippine isolates of *M. oryzae* where 13% were avirulent genotype *Guy11* and 83% were virulent genotypes *PH14* and *CM28* [25].

Most rice farmers in developing nations are financially constrained to purchase chemical inputs to manage rice blast or lack the technical know-how to use such inputs effectively [26]. The use of resistant varieties reduces the environmental risk associated with the dependence on heavy applications of chemical inputs. Genetic resistance is considered the ideal way to control rice blast. However, with a very high genetic variability of the pathogen, each cultivar is useful only for a few years, after which it becomes susceptible to infection as new races of *M. oryzae* develop [27].

This study aimed at assessing the *Avr* genes possessed by the rice blast isolates collected in Philippine rice fields during the 2012 cropping season from various geographic locations and distinct agroecosystems.

## 2. Materials and Mathods

### 2.1. Magnaporthe oryzae Isolates and DNA Extraction

Rice blast isolates used in this study were previously known to represent a distinct genotype, based on Simple Sequence Repeat (microsatellite) analysis made in The French Agricultural Research Centre for International Development (CIRAD), France. These isolates were collected during the 2012 cropping season from various rice-growing areas in the three major islands of the Philippines (Luzon, Visayas and Mindanao) (Figure 1) under different agroecosystems: irrigated lowland, rainfed lowland and rainfed upland. Samplings were made from farmers’ fields as well as from the various experimental stations of the Philippine Rice Research Institute (PhilRice) and those from the blast nursery of the University of the Philippines Los Baños (UPLB).

Monoconidial cultures were maintained in rice flour agar (20 g rice flour, 2 g yeast extract, 15 g agar and 1 L water) combined with 500,000 IU of penicillin G after autoclaving for 20 min. at 120 °C. Total genomic DNA was extracted following the cell-wall digestion procedure as described by Sweigard et al. [28]. Reference strains considered as representative of the diversity were observed in the reference study of Chen et al. [29].

### 2.2. PCR Assay for the Presence of Avirulence Genes

Total genomic DNA was amplified using the primer sets as shown in Table 1. Extracted DNA (10 ng) was used as a template in the PCR reaction containing 10X optimized DyNAzyme Buffer [1X buffer contains: 10 mM Tris-HCl (pH 8.8 at 25°C), 1.5 mM MgCl_2_, 50 mM KCl, 0.1% Triton], 10 µM each of the forward and reverse primers, 10 mM dNTP mix, 0.5 U DyNAzyme II DNA Polymerase (Thermo Scientific, Waltham, MA, USA), and sterile distilled water to a volume of 25 µL. A total of 164 isolates, from 131 newly collected and 33 old and reference isolates also collected from Philippine rice fields but maintained in Didier Tharreau’s laboratory in CIRAD, Montpellier, France was considered.

The PCR program was set at 95 °C for 3 min, followed by 35 cycles of 94 °C for 30 s, 53–64 °C for 45 s, depending on the primer set, and 72 °C for 1 min with a final extension of 72 °C for 7 min in a PTC100 (MJ Research, Waltham, MA, USA) thermocycler. Polymerase chain reaction products were analyzed by 2% agarose gel electrophoresis in TAE buffer for 30 min at 100 V. Gels were stained in ethidium bromide and visualized under UV light.

Multilocus genotypes (MLGs) are a unique combination of alleles across two or more loci and are useful for identifying the spread of the pathogen. Multilocus genotypes were determined through analyzing the result of the DNA sequencing from PCR products amplified utilizing the 12 microsatellite markers. Those belonging to the same MLG have the same sizes of amplified products in all of the 12 microsatellite markers using the software GeneMapper version 4.1 (Applied Biosystems, Foster City, CA, USA). The MLG:isolate ratio means the proportion of unique MLG which can be expressed as percentage from the number of isolates collected.

The determination of the *R* genes present in the rice genotype was mainly attributed to the presence of *Avr* genes. This is based on the gene-for-gene concept of Flor [5] which states that “for each gene that conditions reaction in the host there is a corresponding gene in the parasite that conditions pathogenicity”. Each gene in either member of a host-parasite system may be identified only by its counterpart in the other member of the system. If there are *Avr* genes in the pathogen, the corresponding *R* genes in the host are also assumed to be present.

### 2.3. Data Analysis

Data from plus/minus PCR test to assess for the presence or absence of six *AVR* genes in each isolate were subjected to contingency chi-square (χ^2^) analysis against geographic locations and types of agroecosystem utilizing the JMP-N SAS software v 10 (SAS Institute Inc., Cary, NC, USA). χ^2^ values computed for each cell used the formula (*O − E*)^2^ / *E* where *O* and *E* represent observed and expected frequencies, respectively.

## 3. Results and Discussion

### 3.1. Avirulence Genes in the Isolates

*Avr-Pik* was present in the majority of the isolates (81.50%). The relative frequencies of the other *Avr* genes were 64.2% (*Avr-Pita*), 48.0 % (*Avr-Pii*), 19.1% (*Avr-Pizt*), 8.67% (*Avr-PWL3*) and 5.2% (*Avr-Pia*). Gel electrophoresis showing the migration pattern of the different *Avr* genes is presented in Figure 2. As expected, isolates belonging to the same MLGs possessed similar *Avr* genes except for isolates PH 217 and 358, which differ in *Avr-Pita*. This could not be a potential error as the amplification was done three times. A comparative frequency of *Avr* genes found in isolates collected from each type of agroecosystem and those in the reference strains is shown in Figure 3. The upland agroecosystem was noted to contain all the six classes of avirulence genes, whereas the irrigated lowland areas were devoid of *Avr-Pia* and *Avr-PWL3*, and a very small percentage of isolates from irrigated lowland contain *Avr-Pizt* (Figure 3).

Contingency analysis of the type of agroecosystem and the presence of *Avr* genes in the isolates indicated that there was no significant association between geographic locations (islands) and the presence of *Avr* genes, except for *Avr-PWL3*, which shows to highly prevail in the Visayas (Table 2), but this gene is known to be non-functional in rice [9]. In contrast, *Avr-Pia*, *Avr-Pizt* and *Avr-PWL3* were significantly associated with the upland condition. This seems to show that a specific *Avr* gene would prevail on a certain type of agroecosystem but not on others; however, the distribution of *Avr* gene is affected by the distribution of rice cultivars possessing the corresponding *R* genes. Hence, data on *R* genes present on rice genotypes grown in the various agroecosystems are shown in Table 3.

### 3.2. Resistance Genes Present in Rice Genotypes

Based on the data generated after testing for the presence of *Avr* genes in the isolates, the complementary *R* genes, contained in a rice genotype where a specific *M. oryzae* isolate originated, were thus determined. It was known that the irrigated lowland rice genotypes contained the lowest number of *R* genes (mostly *Pik*, *Pita* and *Pii*), whereas the rainfed upland rice genotypes contained the highest number of *R* genes. Furthermore, the upland rice genotypes locally known as “Milagrosa” and “Waray” from Cavinte, Laguna possessed all of the six complementary *R* genes (Table 3).

Knowledge of the geographic distribution and frequency of avirulence genes present in the collection of isolates will lead to the development of strategies for proper deployment of resistant varieties. Studies dealing with spatial and temporal changes in population diversity of the invading plant pathogen illustrated that *Avr* genes appeared to have a significant degree of diversity at both small and large sampling scales, suggesting that pathogen evolution varies among local conditions [30].

In this study, the ratio of samples amplified is similar for *Avr-Pik*, *Avr-Pita* and *Avr-Pia* to the worldwide samples [31]. *Avr-Pii* was more frequently amplified in our sample but compared to the ratio of the Philippine subsamples (reference strains), it was similar. *Avr-Pizt* was exceptional because it was said to be amplified in 90% of worldwide strains and 67% of the Philippines strains (reference strains), so it seemed that our collected samples were very different. Jia et al. [32] examined the *Avr* genes present on isolates collected in southern US from 1970 to 2009 and reported that the majority of their collections contained *Avr-Pita* (65.7%). The other *Avr* genes and their relative frequencies noted in their collections were *Avr-Pizt* (57.1%), *Avr-Pik* (11.4%), *Avr-Pia* (2.9%) and *Avr-Pii* (0.3%). In Yunnan, China, the presence of *AvR-Pii* was 82 out of 454 field isolates of *M. oryzae* [33], whereas in Eastern India, *Avr-Pizt* and *Avr-Pik* had the highest frequency (100%) and *Avr1-CO39* had the lowest (2%) [34]. In Thailand, *PWL-2*, *Avr-Pii* and *Avr-Pizt* gene-specific primer amplification had the highest frequency of 100%, 60% and 54%, respectively [35]. From this, it can be inferred that there is a different avirulence gene composition of our new isolates, especially for *Avr-Pizt*. *Avr-PWL3* is rarely present in *M. oryzae* isolates from rice [9], but it is even more frequent than *Avr-Pia* in our samples.

Significant correlations were noted on the type of agroecosystem as well as the classes of avirulence genes present. All of the cloned avirulence genes tested were present in fair proportions in the upland agroecosystem. This result reinforces the fact that rice blast is most favorable in the upland [36], and that rainfed still is more favorable than the irrigated lowland. In spite of this, examining the data further had made known that *Avr-Pik* and *Avr-Pii* occur in highest frequency in the irrigated lowland and rainfed lowland, respectively. This implies that the rice varieties planted can have significant impact on the existence of avirulence genes in the area. This result corroborates with those of Li et al. [37] who studied the geographic distribution of avirulence genes in rice blast fungus in Yunnan province in China. Their study illustrated that the composition and distribution of rice genetic diversity are more important than climate and environmental conditions in the prevalence of avirulence genes in populations of *M. oryzae*. Considering the weak geographic structure of *M. oryzae* populations in the Philippines, where gene flow occurs among the major islands, usually through the transport of infected seeds [38], the spread of avirulence genes as well could impact the stability of resistance genes in the agricultural fields. Some avirulence genes tested in this study are functional, such as *Avr-Pizt* and *AvrPii*, based on our results of pathogenicity test [38]. However, the polymorphisms of the avirulence genes were not tested, and the pathotypes of all the 131 isolates were not completely investigated. Therefore, it is advisable to conduct these experiments for future outlook of the project. Such strategy of assessing the avirulence genes currently present in an area would be useful for efficient deployment of resistant genes in order to better manage rice blast.

## Figures and Tables

**Figure 1 microorganisms-07-00023-f001:**
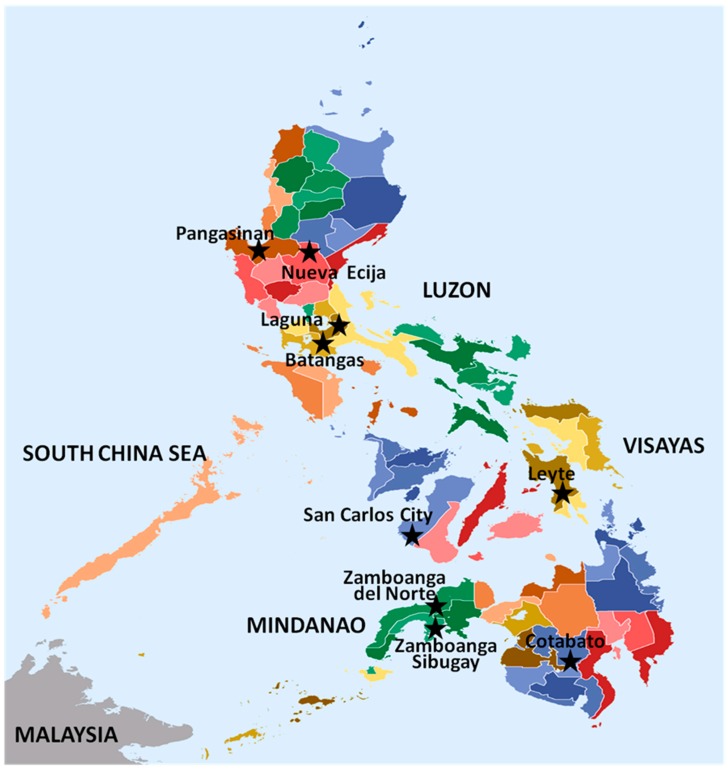
A map of the Philippines showing sampling locations of *Magnaporthe oryzae* isolates from various rice growing areas. Actual locations are named and indicated with a 5-point star shape.

**Figure 2 microorganisms-07-00023-f002:**
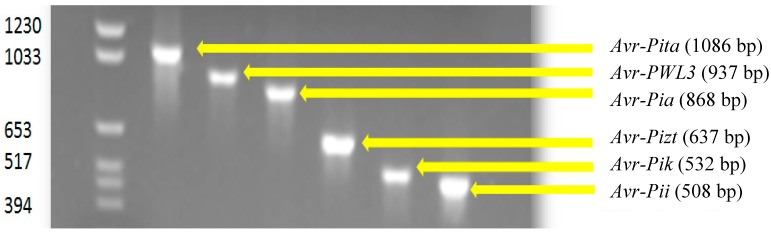
Agarose gel electrophoresis of PCR products from *M. oryzae* isolates tested for a specific avirulence gene.

**Figure 3 microorganisms-07-00023-f003:**
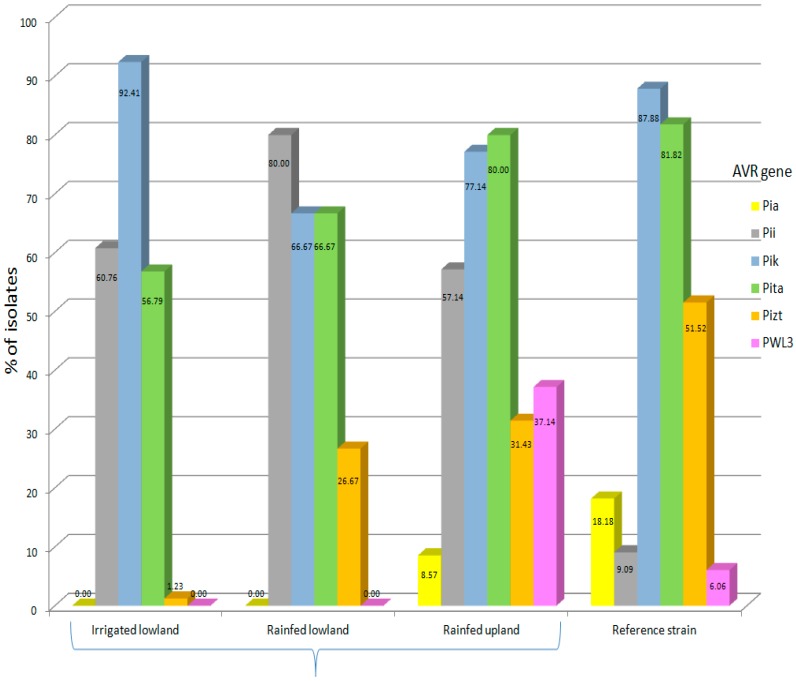
Comparative frequency of avirulence genes in the contemporary *M. oryzae* genotypes collected across the various agroecosystems and in the reference strains. Sample sizes for irrigated lowland (90), rainfed lowland (15), rainfed upland (35) and reference strains (33).

**Table 1 microorganisms-07-00023-t001:** Specific sequences, avirulence gene amplified and PCR product size of primers tested against *M. oryzae* isolates.

Primer Name	Sequence 5’–3’	*AVR* Gene	Expected Product Size (bp)
YL169b	CGACCCGTTTCCGCC	*Avr-Pita*	1086
YL149b	TGACCGCGATTCCCTCCATT		
Z21	AATCCCGTCACTTTCATTCTCCA	*Avr-Pizt*	637
Z22	GTCGCAAGCCTCGTACTACCTTT		
Z23	TCCAATTTATTCAACTGCCACTC	*Avr-Pik*	532
Z24	GTAAACCTCGTCAAACCTCCCTA		
Z27	CCCATTATCTTACCAGTCGCTTGA	*Avr-Pia*	868
Z28	ATTCCTCCCGTAAACAGTAAACC		
ZZ41	TGCAGGCCCAAATCCGTAGGAA	*Avr-Pii*	508
ZZ42	ACTGTCCGCCGCTCGTTTGG		
PW3-F	TGCGTGCTCATTTGTAAACC	*Avr-PWL3*	937
PW3-R	TTGACGGTACTAGGGCTGCT		

**Table 2 microorganisms-07-00023-t002:** Expected frequency (*E*) and computed chi-square (χ^2^) values for the presence (1) and absence (0) of a specific avirulence gene in *M. oryzae* isolates against geographic location and type of agroecosystem.

	χ^2^ Value*											
Geographical Location/Type of Agroecosystem	*Avr-Pita*		*Avr-Pia*		*Avr-Pizt*		*Avr-Pik*		*Avr-Pii*		*Avr-PWL3*	
	0	1	0	1	0	1	0	1	0	1	0	1
**A. Geographic Location**												
Luzon												
*E*	28.60	53.39	80.09	1.91	71.83	10.17	12.07	69.92	31.15	50.86	73.74	8.26
χ^2^	0.74	0.40	0.02	0.63	0.37	2.63	0.07	0.01	0.04	0.03	0.25	2.20
Mindanao												
*E*	6.98	13.02	19.53	0.47	17.52	2.48	2.95	17.05	7.60	12.40	17.98	2.02
χ^2^	7.07	3.79	0.01	0.46	0.36	2.56	0.30	0.05	0.76	0.47	0.23	2.02
Visayas												
*E*	9.42	17.58	26.37	0.63	23.65	3.34	3.97	23.02	10.26	16.74	24.28	2.72
χ^2^	0.62	0.33	0.02	0.63	0.30	2.10	0.00	0.00	0.15	0.09	1.62	14.92
**B. Type of Agroecosystem**												
Irrigated lowland												
*E*	27.56	51.44	77.16	1.84	69.20	9.79	11.64	67.36	39.24	60.76	71.04	7.96
χ^2^	1.075	0.58	0.04	1.84	1.12	7.90	2.73	0.47	0.03	0.02	0.89	7.96
Rainfed lowland												
*E*	5.23	9.77	14.65	0.35	13.14	1.86	2.21	12.79	20.00	80.00	13.49	1.51
χ^2^	0.01	0.01	0.01	0.35	0.35	2.46	3.53	0.61	1.28	0.78	0.17	1.15
Rainfed upland												
*E*	12.21	22.79	34.19	0.82	30.66	4.34	5.16	29.84	42.86	57.14	31.47	3.53
χ^2^	2.22	1.19	0.14	5.87	1.45	10.21	1.57	0.27	0.22	0.13	2.85	25.44

* Color in orange; χ^2^ values indicate significant association between a specific avirulence gene and geographic location or type of agroecosystem.

**Table 3 microorganisms-07-00023-t003:** Resistance genes identified through PCR assay of *M. oryzae* isolates taken from each of the host genotypes in distinct agroecosystems.

Host Genotype	Resistance Gene *					
	*Pita*	*Pia*	*Pizt*	*Pik*	*Pii*	*PWL3*
**A. Irrigated Lowland**						
012–N	0	0	0	1	1	0
15 IL	1	0	0	1	1	0
Au 108	1	0	0	1	1	0
Au 63	1	0	0	1	1	0
Bigante	0	0	0	1	1	0
C9305-B-9-2	0	0	0	1	0	0
Dinorado	1	0	0	1	0	0
ILMAS 540	1	0	0	1	1	0
IR 50	1	0	0	1	1	0
IR83140-B-36-B	0	0	0	1	0	0
IR86781-3-3-1-1	1	0	1	1	1	0
LMAS 544	1	0	0	1	1	0
MS 11	0	0	0	1	0	0
NSIC Rc130	1	0	0	1	0	0
NSIC Rc212	0	0	0	1	1	0
NSIC Rc216	1	0	0	1	1	0
PNG 719	1	0	0	1	1	0
PR34350-4-POKKALI-24-M5R-10	0	0	0	1	1	0
PR36723-B-13-3-3-3	0	0	0	1	0	0
PR36930-B-7-3	0	0	0	1	0	0
PR37088-B-9-1-1	0	0	0	1	0	0
PR37624-1-5-1-2-1	0	0	0	1	0	0
PR37801-15-1-1-3-2-B-B	0	0	0	0	1	0
PYT 132	1	0	0	1	0	0
PYT 172	1	0	0	1	1	0
PYT 20	1	0	0	1	1	0
PYT 210	1	0	0	1	1	0
PYT 218	1	0	0	1	0	0
PYT 238	1	0	0	1	0	0
PYT 307	1	0	0	1	1	0
PYT 324	1	0	0	0	1	0
PYT 41	1	0	0	1	1	0
**B. Rainfed Lowland**						
Bigante	1	0	1	1	1	0
C4	0	0	0	0	1	0
NSIC Rc222	1	0	0	1	1	0
**C. Rainfed upland**						
Japanese	1	0	1	1	0	1
Milagrosa	1	1	1	1	1	1
NSIC Rc152	1	0	1	1	1	1
NSIC Rc216	1	0	0	1	1	0
PhilRice M3	1	0	1	1	1	1
Waray	1	1	1	1	1	1

* 0 = absence of a specific resistance gene; 1 = presence of a specific resistance gene.

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
