# Peer review of "Geographic Distribution of Avirulence Genes of the Rice Blast Fungus Magnaporthe oryzae in the Philippines"

_microorganisms, 2019, doi:10.3390/microorganisms7010023_

Round 1
Reviewer 1 Report
This manuscript investigated the geographic distribution of avirulence genes in rice blast fungus in Philippines. This study would contribute to the development of strategies to use rice varieties to improve rice blast disease control. But there are a few experiments required to be carried out before the publication of this manuscript.
1. The authors listed many avirulence genes in rice blast fungus in the introduction (P1, L43), but why only a few of them were studied in this manuscript?
2. Do the authors see any polymorphisms of the avirulence genes?
3. Are these avirulence genes from the collected isolates functional? Are there any pathogenicity assays to prove this? If some of the avirulence genes are not functional, the results of this study will mislead the deployment of effective rice cultivars.
Author Response
Reviewer 1
This manuscript investigated the geographic distribution of avirulence genes in rice blast fungus in Philippines. This study would contribute to the development of strategies to use rice varieties to improve rice blast disease control. But there are a few experiments required to be carried out before the publication of this manuscript.
1. The authors listed many avirulence genes in rice blast fungus in the introduction (P1, L43), but why only a few of them were studied in this manuscript?
We tested almost all of them (Avr-Pita, Avr-PWL3, Avr-Pia, Avr-Pizt, Avr-Pik, Avr-Pii) except Avr1-CO39. For the ACE1, we already published the result with this article:
Lopez, A L. C., Cumagun, C.J.R. and Didier, T. 2015. Diversity of ACE1 genotypes of the rice blast fungus (Magnaporthe oryzae B.C. Couch) in the Philippines. IAMURE International Journal of Ecology and Conservation 16:80-92.
2. Do the authors see any polymorphisms of the avirulence genes?
We did not test for any polymorphisms of the avirulence genes.
3. Are these avirulence genes from the collected isolates functional? Are there any pathogenicity assays to prove this? If some of the avirulence genes are not functional, the results of this study will mislead the deployment of effective rice cultivars.
We published already the pathogenicity test of 24 isolates with different avirulence genes (Table 8) in this article:
Lopez, ALC. and Cumagun, C.J.R.2018. Genetic structure of Magnaporthe oryzae populations in three island groups in the Philippines. European Journal of Plant Pathology https://doi.org/10.1007/s10658-018-1546-0
Based on this results, Avr-Pizt and AvrPii are functional as mentioned in Lines 209-210.
Table 8 Reactions of the differential rice varieties to contemporary genotypes of Magnaporthe oryzae after pathogenicity test.
Genotype/ Isolate | Rice Varieties (R genes)* | |||||||||||||||
A | B | C | D | E | F | G | H | I | J | K | L | M | N | O | P | |
PH0200 | R | R | R | R | R | R | R | R | R | R | R | R | R | R | HS | R |
PH0201 | R | HS | R | R | R | R | HS | R | R | R | R | HS | R | HS | R | HS |
PH0210 | HS | HS | R | R | R | R | HS | R | R | MS | R | MS | MS | R | R | R |
PH0211 | ND | R | R | R | R | MS | R | HS | R | R | R | R | R | R | R | HS |
PH0213 | HS | HS | R | R | R | R | HS | HS | R | R | HS | ND | R | R | R | HS |
PH0241 | R | HS | R | R | MS | R | HS | R | R | R | R | HS | R | R | R | HS |
PH0242 | R | HS | R | R | MS | R | HS | R | R | R | R | MS | R | R | R | HS |
PH0248 | R | HS | R | R | MS | R | HS | R | R | R | R | HS | R | R | R | HS |
PH0278 | HS | HS | R | R | R | R | MS | HS | R | R | HS | R | R | R | R | HS |
PH0281 | R | HS | R | MS | R | R | MS | HS | R | R | HS | R | R | R | R | HS |
PH0304 | R | HS | R | R | R | R | MS | HS | R | R | HS | R | R | R | R | HS |
PH0318 | R | HS | R | R | MS | R | HS | R | R | R | R | HS | R | R | R | HS |
PH0324 | R | HS | R | R | MS | R | HS | R | R | R | R | HS | R | R | R | HS |
PH0326 | R | HS | R | R | MS | R | HS | R | R | R | R | HS | R | R | R | HS |
PH0328 | ND | HS | R | R | R | R | HS | HS | R | R | HS | HS | R | R | R | HS |
PH0329 | MS | HS | R | R | MS | R | HS | HS | R | R | HS | HS | R | HS | R | HS |
PH0349 | R | HS | R | R | MS | R | HS | R | R | R | R | HS | R | R | R | HS |
PH0351 | MS | HS | R | R | R | R | HS | R | R | R | R | HS | R | HS | R | HS |
PH0358 | R | HS | R | R | R | R | HS | R | R | R | R | HS | R | R | R | HS |
PH0361 | R | HS | R | R | R | R | HS | R | R | R | R | MS | R | R | R | HS |
PH0383 | R | HS | R | R | MS | R | HS | R | R | R | R | HS | R | R | R | HS |
PH0384 | R | HS | R | R | MS | R | HS | R | R | R | R | HS | R | R | R | HS |
PH0427 | R | HS | R | MS | R | R | HS | HS | R | R | HS | HS | R | R | R | HS |
PH0432 | R | HS | R | R | ND | R | HS | HS | R | R | HS | HS | R | R | R | HS |
R = Resistant; MS= Moderately susceptible; HS = Highly susceptible; ND = not determined
* A = Azucena (Pi24);B = CO39 (PiCO39);C = C101Lac (PiCO39, Pi1, Pi33);D = Fukunishiki (Piz, Pish);E = C101A51 (PiCO39, Pi2);F = Tsuyuake (Pikm);G = C101TTP (PiCO39, Pita);H = Fujisaka 5 (Pii, Piks);I = C104Lac (PiCO39, Pi1);J = 75-1-127 (Pi9);K = C104PKT (PiCO39, Pi3);L = Toride 1(Pizt);M = IR1529 (Pi33);N = Bala (Pi33);O = IR64 (Pi33);P = Maratelli
Reviewer 2 Report
In this paper, the authors investigated the effect of geographic location and agroecosystem on the appearance of Avr genes in Magnaporthe oryzae strains and R gene in rice host. And showed that a significant increases of AvrPia, AvrPizt, and AvrPWL3 in the M. oryzae strains isolated from upland, whereas the rice resistance genes Pia, Pizt, and PWL3 appeared more in the upland rice genotypes.
In general, the data is very clear, conclusions are justified, and the manuscript is easy to follow. However, still some part need to be revised before publish.
1. Recognition between R and Avr genes causes hypersensitive resistance in rice, which kills the infected fungus in host. The authors should explain the meaning of why this high ratio of R-Avr pairs are identified in the uplands. Following this point, in the abstract part, the authors claims that ‘It was also laid out that the presence of Avr genes in M. oryzae is strongly associated with agroecosystems where the complementary R genes exist.’ Since they did not test whether those M. oryzae strains could really infect those host genotypes. The authors should consider to weaker this claim.
2. Line 148, a Table 4.2 appears in the manuscript, but no table 4.2 appears. The authors should check.
Author Response
Reviewer 2
In this paper, the authors investigated the effect of geographic location and agroecosystem on the appearance of Avr genes in Magnaporthe oryzae strains and R gene in rice host. And showed that a significant increases of AvrPia, AvrPizt, and AvrPWL3 in the M. oryzae strains isolated from upland, whereas the rice resistance genes Pia, Pizt, and PWL3 appeared more in the upland rice genotypes.
In general, the data is very clear, conclusions are justified, and the manuscript is easy to follow. However, still some part need to be revised before publish.
1. Recognition between R and Avr genes causes hypersensitive resistance in rice, which kills the infected fungus in host. The authors should explain the meaning of why this high ratio of R-Avr pairs are identified in the uplands. Following this point, in the abstract part, the authors claims that ‘It was also laid out that the presence of Avr genes in M. oryzae is strongly associated with agroecosystems where the complementary R genes exist.’ Since they did not test whether those M. oryzae strains could really infect those host genotypes. The authors should consider to weaker this claim.
There is a high ratio of R-Avr pairs are identified in the uplands because most of the R genes are found in upland rice varieties (see Lines 139, 167-168). The upland rice varieties in our study possessed all complementary R genes. Again we published the pathogenicity test in Lopez and Cumagun, European Journal of Plant Pathology 2018.
2. Line 148, a Table 4.2 appears in the manuscript, but no table 4.2 appears. The authors should check.
We deleted Table 4. 2 which is not included in the paper.
Round 2
Reviewer 1 Report
It’s very important to make sure all the avirulence genes present in different isolates in this study are functional or not functional. This could be done by simple pathogenicity test or polymorphism analysis. If the authors are not going to do these experiments in this study, it’s better to specify the results they got in this manuscript. For example, the polymorphisms are not tested, and the pathotype of all the 131 isolates are not completely investigated. This manuscript can be considered for publication as long as it comes with proper description of the results.
Author Response
Reviewer 1
Comment
It’s very important to make sure all the avirulence genes present in different isolates in this study are functional or not functional. This could be done by simple pathogenicity test or polymorphism analysis. If the authors are not going to do these experiments in this study, it’s better to specify the results they got in this manuscript. For example, the polymorphisms are not tested, and the pathotype of all the 131 isolates are not completely investigated. This manuscript can be considered for publication as long as it comes with proper description of the results.
Response
We wrote the following under Discussion on page 9, line 211-215:
However, the polymorphisms of the avirulence genes were not tested, and the pathotypes of all the 131 isolates were not completely investigated. Therefore, it is recommended to conduct these experiments for future outlook of the project. Such strategy of assessing the avirulence genes currently present in an area would be useful for efficient deployment of resistant genes in order to better manage rice blast.